# Progress towards an HF Radar Wind Speed Measurement Method Using Machine Learning

Lucy R. Wyatt [1,2]

1 Seaview Sensing Ltd., Sheffield S10 3GR, UK; lucywyatt@seaviewsensing.com or l.wyatt@sheffield.ac.uk
2 School of Mathematics and Statistics, University of Sheffield, Sheffield S10 2TN, UK

**Abstract:** HF radars are now an important part of operational coastal observing systems where they are used primarily for measuring surface currents. Their use for wave and wind direction measurement has also been demonstrated. These measurements are based on physical models of radar backscatter from the ocean surface described in terms of its ocean wave directional spectrum and the influence thereon of the surface current. Although this spectrum contains information about the local wind that is generating the wind sea part of the spectrum, it also includes spectral components propagating into the local area having been generated by winds away from the area i.e., swell. In addition, the relationship between the local wind sea and wind speed depends on fetch and duration. Thus, finding a physical model to extract wind speed from the radar signal is not straightforward. In this paper, methods that have been proposed to date will be briefly reviewed and an alternative approach is developed using machine learning methods. These have been applied to three different data sets using different radar systems in different locations. The results presented here are encouraging and proposals for further development are outlined.

**Keywords:** wind speed; HF radar; machine learning; wind direction; support vector machine; regression; coastal monitoring; marine; ocean

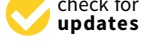

## 1. Introduction

HF radar systems [1] are routinely used in many parts of the globe to provide continuous, spatial monitoring of surface currents in the coastal ocean [2–5]. They provide important data for applications such as pollutant dispersal, search and rescue, and for understanding oceanographic and ecosystem processes. The ocean wave directional spectrum and associated parameters, e.g., significant waveheight, peak period and direction, can also be obtained from the radar signals; see [6] for an overview. The availability of wave measurements does vary with the type of radar used and its operating frequency and is more limited in range from the coast than current measurements, but an operational capability has been demonstrated [7,8]. Wind direction measurements are also available and have been extensively validated [7,9,10]. A robust measurement of wind speed has proved to be more illusive but would be a very useful addition to the measurement capability of these systems with important application to offshore wind farm resource assessment, installation and performance monitoring [11].

Wave and current measurements are based on physical models of radar backscatter from the ocean surface described in terms of its directional spectrum and the influence thereon of the surface current [12–14]. They use the Doppler spectrum of the backscatter which can be described in terms of first and second order contributions (see Figure 1). Although this spectrum contains information about the local wind, which is generating the wind sea part of the spectrum, it also includes impacts from spectral components propagating into the local area having been generated by winds away from the area i.e., swell. In addition, the relationship between the local wind sea and wind speed depends on fetch and duration. Thus, finding a physical model to extract wind speed from the radar

signal is not straightforward. A recent overview of HF radar wind speed and direction measurement [15] reviews a number of attempts that have been made to use both the first order and second order parts of the Doppler spectrum to obtain wind speed but none of these have yet produced a robust estimate with an accuracy useful for operational and scientific applications. One method referred to in that review that uses the second order spectrum is an inverse wind wave modelling method [16]. This is implemented in the Seaview Sensing Ltd. operational software, but we know it to be unreliable in the presence of swell and therefore in particular in low winds. This is evident in the accuracy assessment of the method given in Table 1. Validations in more wind sea dominated situations have been more successful. Other methods discussed in [15], for which published accuracies are also included, are based only on the first order part of the spectrum and include applications of Neural Networks (NN), partial least squares (PLS) and physical models of the first order amplitudes. An NN approach using metocean parameters: significant waveheight, mean period and wind direction is presented in [17]. This was trained using buoy data and then tested with data from a 13 MHz Seasonde radar. These results are also included in Table 1. The rms values quoted need to be judged with respect to the range of wind speeds in the comparison period. An rms of 1 ms$^{-1}$ in a period of low winds may be no more accurate than 2 ms$^{-1}$ if a wider range of wind speeds has been included in the analysis. A scatter index (SI = rms/mean) would be a more useful measure, but this was not provided for most of these data sets. An additional scatter index (SI$_{max}$ = rms/max) is therefore included in the table to provide a comparator for the new results from this paper. This is not as robust a measure since the maximum wind speed is not usually representative of the variability in the measured data. Note that a standard error of prediction (sep) was given in the PLS case [18] instead of an rms, so SI$_{max}$ has not been calculated.

**Table 1.** Published wind speed accuracies.

| Source | Method | rms/sep ms$^{-1}$ | SI | SI$_{max}$ | Bias ms$^{-1}$ | Correlation Coeff | Max Speed ms$^{-1}$ |
|--------|--------|------|----|-----|------|-------------|-----------|
| [7] | Inverse wave model [16] | 5.188 | | 0.35 | | | 15.0 |
| [18] | PLS | 1 | | | −0.4 | 0.8 | 14 |
| [19] | NN using peaks | 2.07 | | 0.1 | | 0.84 | 20 |
| [19] | NN using spreading | 2.8 | | 0.14 | | 0.67 | 20 |
| [15] | First-order model [20] | 1.89 | | 0.19 | | | 10 |
| [17] | NN using metocean | 1.7 | 0.34 | 0.13 | 1.37 | 0.68 | 13 |

In this paper, we present a new approach using machine learning methods applied to various features extracted from the radar Doppler spectrum together with some of the derived metocean measurements. Machine learning methods really need co-located, co-temporal wind speed data in order to train models, but these are not always available. In this paper, we use data from an in situ mast, a coastal wind measurement close to a radar site rather than the offshore measurement position, and UK Met Office model wind data. These are from three radar sites on the UK coast, at Liverpool Bay and in the southern and northern parts of the Celtic Sea, respectively.

We note that the accuracy requirements (compared with buoy measurements) for satellite scatterometer wind speeds are bias < 0.5 ms$^{-1}$ and standard deviation < 2 ms$^{-1}$ [21]. The latest results [22] show biases of up to about 0.2 ms$^{-1}$ and standard deviations of 1.2–2.0 ms$^{-1}$ when compared with buoys. The Carbon Trust [23] requires a linear regression slope of 0.98–1.02 and R$^2$ > 0.98 for wind speed measured with LiDAR. Our eventual goal is to at least match if not better scatterometer accuracies evaluated over a wide range of wind speeds. This paper represents a step in that direction.

In Section 2, the methods that have been tested and the data sets to which they have been applied are discussed. The results are presented in Section 3 followed by a discussion in Section 4 including proposals for further work.

## 2. Materials and Methods

### 2.1. Data Sets

Three different UK data sets are being used to develop and test the methods. These are summarised in Table 2.

The first two data sets used FMCW (frequency-modulated continuous wave) WERA radars, the frequencies of which can be varied over narrow bands to limit interference. WERA radar measurements are provided on a rectangular grid with a cell size determined by the range resolution.

The WERA radars in Liverpool Bay (LB) were operated by NOCL and installed and maintained by Neptune Radar Ltd. These radars were part of the Liverpool Bay coastal observatory [24]. The radar measurements have a cell size of 4 km. In this case, NPower Renewables (now part of E.ON) have provided several months of wind data from a mast located in the field of view of the radars at a height of 25 m. A simple log scaling has been applied to provide winds at 10 m so that the radar measurement will be nominally at this height. A condition of supply is that these wind speed data have to be normalised if they are displayed and that has been done in the work presented here. Radar data from the cell containing the mast have been used. Although this is the only data set involving offshore wind measurements, and therefore the best data set for this type of work, the radar operating frequency (see Table 2) proved to be rather low for the prevailing wave conditions limiting the the number of cases with sufficient signal-to-noise for this analysis. Four months of data have been used split 50:50 into training and testing data sets,

The south Celtic Sea radars are operated by the University of Plymouth [25]. These have a cell size of 1 km. There were no offshore wind measurements in this case, so measurements from the coastal station at Perranporth [26] have been used for this paper. Quality controlled data were downloaded and are available at 10 min intervals. Dual and single radar data from two cells are used separately in the analyses presented here located at distances of 13.6 (SCS1) and 16.7 km (SCS2) from the coastal station. Five weeks of data were available for this analysis and were split 90:10 into training and test data sets. Using fewer data in the training set in order to increase the amount of data available for testing did not give satisfactory results because the range of metocean conditions was not wide enough.

The Pisces radars were installed and operated by Neptune Radar Ltd. for an operational trial to establish the availability and accuracy of wave data at a site 60 km from the coasts of North Devon and South Wales [7] (NCS). These are FMICW (I - interrupted) radars operating over a range of frequencies (see Table 2) adjusted according to environmental conditions. For this trial, they were adjusted to minimise interference, but they could also be adjusted to maximise wave data return in different sea-states [27]. A wave buoy was located at the 60 km position to validate the radar wave measurements, but there were no offshore wind measurements during this trial. The UK Met Office provided model wind data and also compared the radar measurements with QuickSCAT scatterometer data [7], which confirmed the limitations of the radar measurement in low wind, swell-dominated conditions (see Table 1). The model data are used in this paper for the machine learning and model and QuickSCAT are used in the validation. In this trial, radar measurements were made sequentially for 19 minutes on each of three beams for each radar every hour. Single radar measurements [28] were made along these beams with a range resolution of 15 km. This configuration provides nine beam intersections between the radars at each of which the two nearest range-azimuth cells from each radar are combined with each of those from the other radars to provide four separate dual metocean data sets (wave, current, and wind) geo-located to the mean of the positions of the centres of the radar cells used. The nine beam intersections are time stamped with the start time of the measurement from the North Devon site. There are thus a maximum of 36 measurements each hour across a region extending to more than 150 km from the coast, although very few measurements were obtained at the furthest intersection point (more than 200 km from both radars), where signal to noise from both sites is too low. A threshold of 8 dB is used in this analysis. Data

from 8 of these points are used in the analysis; the rest are used for further validation. The nearest model wind speed and directions to each of these locations were used in this analysis, noting that the model resolution was 12 km, which is comparable to the radar range resolution. Spatial separations are between 0.5 and 8 km. Model data were available at 6 hourly intervals: 0, 6, 12, 18 h. Given the coarse radar spatial and temporal configuration, it was not thought necessary to interpolate the model data in either time or space in this work. Eighteen months of data were collected and split 50:50 into training and testing data sets.

**Table 2.** Radar, training and validation data sets.

| Location | Sites | Radar | Dates | Frequency MHz | Wind 'Truth' |
|---|---|---|---|---|---|
| Liverpool Bay (LB) | Llandulais Formby | WERA | October 2005– February 2006 | 12.45–13.43 | In Situ anemometer |
| S Celtic Sea (SCS) | Perranporth Pendeen | WERA | November 2012 | 11.77–12.43 | Perranporth Coastal |
| N Celtic Sea (NCS) | Nabor Point Castlemartin | Pisces | December 2003– June 2005 | 5.73–10.43 | Met Office Model |

*2.2. Methods*

The analysis involves five steps:

1.   Identify radar features/metocean parameters that may be linked to wind speed.
2.   Apply machine learning (ML) methods using these features for a subset of the data (training set) and the available wind product to develop the wind speed measurement model
3.   Test the model on a different subset of the data (testing set).
4.   Evaluate the results and select the best ML method for these data.
5.   Apply the best model to data sets not included in the above to determine whether the method is robust to changes in location, frequency, depth, etc.

In this section, the first two steps are discussed. The remaining steps are discussed in Section 3.

2.2.1. Radar Parameters

At this exploratory stage, a number of features of the radar power (Doppler) spectra and metocean parameters that could be related to wind speed or have an impact on the relationship with wind speed have been identified and all are included in the analysis. They are listed together with the reason for inclusion in Table 3. They are not all independent— for example, wind direction and directional spreading are derived from the Bragg ratios assuming a short-wave directional model [9]. The Doppler spectra features are shown in Figure 1. The Doppler spectra features and radial currents are calculated for each radar separately, the other metocean parameters combine data from the two radars that comprise the radar system at each site. The LB and SCS analyses used data at a single cell with a fixed depth (ignoring any tidal impacts), very small frequency variations and in two fixed directions to the individual radar sites (beam direction), so, in these cases, depth, frequency and radial direction are not used. To ensure some link between the features and wind speed, $U_w$, the radar data are only used if the first order Bragg peak frequencies, $f_b = \frac{\sqrt{4\pi g f_R/300}}{2\pi}$ (where $g$ is gravity, $f_R$ radar frequency in MHz) are less than the peak frequency, $f_{PM}$, given by a Pierson–Moskowitz model (see, e.g., [29]) $f_{PM} = \frac{1}{0.729U_w}$. If this condition is not satisfied, the second-order Doppler signal is likely to be dominated by swell. Because the individual radars may be operating at different frequencies, the average frequency is used in this filtering step.

The variability in the relationship between wind speed and the potentially dependent features and parameters both across the different data sets and for the different radars at each site is clear in Table 4, which shows their correlation coefficients. Although some

correlation coefficients are quite small, all the variables have been retained. The importance of the features to the resulting regressions are discussed in Section 4.

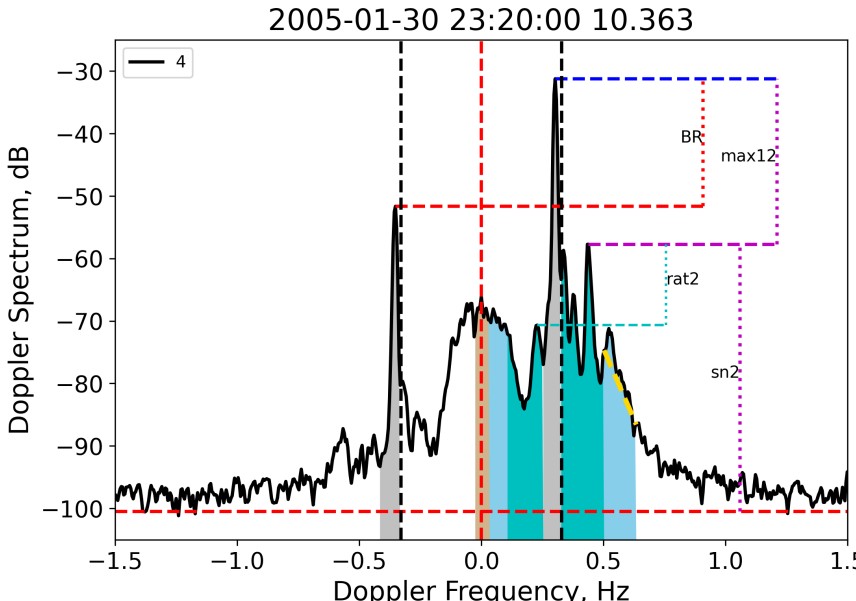

**Figure 1.** Sample Pisces Doppler spectrum (limited Doppler frequency range shown for clarity) showing features used. Lower red dashed line is the noise level. Vertical black dashed lines mark the position of the peaks with no current at this frequency. The two Bragg peaks are grey-shaded and marked with blue and red horizontal dashed lines and are displaced by a current component away from the radar; 2nd order peaks shown with magenta and cyan dashed lines; 2nd order region used for waveheight estimation in cyan, extension for period in light blue. The centre region is in brown. The yellow dashed line provides the slope parameter.

**Table 3.** Doppler spectra features and metocean parameters used in the ML. RI is either R—feature could be related to wind speed or I—feature could impact on wind speed relationships of other features.

| Feature | RI | Reason |
|---|---|---|
| Bragg ratio (BR) | I | As wind speed increase the power in the spectrum increases in a way that depends on wind direction which in turn is related to the Bragg ratio. |
| integrated central region | R | Signal from short wind waves [30] |
| integrated second order | R | Used in empirical waveheight estimates and in part due to local wind |
| integrated second order moment | R | Used in empirical period estimate |
| 2nd order ratio (rat2) | R | Depends on wave directions and amplitudes |
| 2nd order peak signal to noise (sn2) | R | Increases with increasing wind speed |
| 2nd order relative to 1st order (max12) | R | Decreases with increasing wind speed |
| 2nd order slope | R | Reduces as wind speed increases[31] |
| water depth | I | 2nd order power for same wave conditions varies with water depth |
| radar frequency | I | as for depth |
| radial current speed | R | Part of current may be related to wind speed |
| radial (or beam) direction | I | Influences the radial current speed |
| vector current speed, u- v-components | R | as above |
| wind direction | I | as for Bragg ratio |
| short wave directional spreading | R | Empirical evidence from many sources that this is linked to wind speed e.g., [29] |

**Table 4.** Correlation coefficients between Doppler spectra features and metocean parameters labelled R in Table 3 and wind speed for the training data sets. Where two figures are given, the parameters refer to each radar.

| Feature | LB | SCS1 | SCS2 | NCS |
|---|---|---|---|---|
| integrated central region | 0.01/0.03 | 0.36/−0.21 | 0.29/−0.02 | 0.21/0.04 |
| integrated second order | 0.53/0.63 | 0.25/0.48 | 0.25/0.31 | 0.38/0.48 |
| integrated second order moment | 0.47/0.5 | −0.01/0.05 | −0.05/−0.05 | 0.05/0.23 |
| rat2 | 0.01/−0.28 | −0.14/0.13 | −0.07/−0.14 | 0.09/0.12 |
| sn2 | 0.37/0.59 | −0.01/0.37 | 0.06/0.23 | 0.19/0.41 |
| max12 | −0.37/−0.57 | −0.06/−0.49 | −0.19/−0.3 | −0.35/−0.5 |
| 2nd order slope | 0.44/0.45 | −0.06/−0.02 | 0.0/−0.06 | 0.3/0.37 |
| radial current speed | −0.1/−0.06 | −0.04/−0.28 | 0.0/−0.11 | −0.11/−0.09 |
| vector current speed | −0.16 | 0.03 | 0.1 | 0.1 |
| u | 0.07 | 0.08 | 0.02 | 0.12 |
| v | −0.19 | −0.16 | −0.16 | −0.02 |
| short wave directional spreading | 0.45 | 0.29 | 0.23 | 0.33 |

### 2.2.2. Machine Learning Methods

Machine Learning (ML) methods available in the Python Sklearn package [32] are used in this work. The package also includes linear regression methods and some of these (Linear, Lasso, Ridge, Bayesian Ridge, Elastic Net) have also been applied, although with limited success, confirming that the relationship between radar parameters and wind speed is not linear. In addition to the linear methods, nonlinear, partial least squares, support vector machine (SVM) and randomised tree regression methods are tested. SVM regression (SVR) provided the most robust wind speed estimates and most of the results in Section 3 refer to this method, so it is outlined here. For more details on the other methods used, the reader is referred to the Sklearn documentation [32] and to the associated website [33]. A useful tutorial on SVR is given in [34].

In standard linear regression, the aim is to find the coefficients, $w, b$, of a linear model, $y = wx + b$ that minimise the errors of fitting the model to the data, $y_i, X_i, i = 1 \ldots N$, as measured by the square of the L2 norm ($||.||$) of the errors.

$$\min_{w,b} \quad ||\langle wX \rangle + b - y||^2, \tag{1}$$

where $\langle . \rangle$ is the inner product.

In SVR linear regression, the aim is to minimise the L2 norm of the weights in a linear model subject to the errors being within specified bounds. Ref. [34] show how this idea can be extended to provide a nonlinear algorithm leading to the basic SVR method implemented in Sklearn. The nonlinear problem can thus be expressed as

$$\min_{w,b,\zeta_+,\zeta_-} \frac{1}{2}||w||^2 + C \sum_{i=1}^{N} \zeta_{-i} + \zeta_{+i}, \tag{2}$$

subject to

$$y_i - \langle w\Phi(X) \rangle - b <= \epsilon + \zeta_{+i}, \tag{3}$$

$$\langle w\Phi \rangle + b - y_i <= \epsilon + \zeta_{-i}, \tag{4}$$

$$\zeta_{-i}, \zeta_{+i} >= 0. \tag{5}$$

Here, $C$ and $\epsilon$ are selectable parameters, and $\Phi$ is an unknown nonlinear function. As explained in [34], a dual problem is solved that avoids the need to determine $\Phi$, which

is replaced by a kernel function $k(X, X') = \langle \Phi(X)\Phi(X') \rangle$ various options for which are included in Sklearn. Here, the radial basis function is used:

$$k(X, X') = exp(-\gamma ||X - X'||^2), \tag{6}$$

which introduces a third selectable parameter, $\gamma$. Sklearn recommend using their Grid-SearchCV module to determine best values of $C, \epsilon$ and $\gamma$. This was done for the Liverpool Bay data set, giving values the same as the Sklearn SVR defaults of $C = 1, \epsilon = 0.1, \gamma =$'scale'.

The performance of the training and predictions are measured using a number of different metrics where $y$ is the measured wind speed, $y_{std}$ its standard deviation, and $x$ is wind speed derived from the model:

$$bias = \overline{y - x}, \tag{7}$$

$$\text{Mean Absolute Error, } MAE = \overline{|y - x|}, \tag{8}$$

$$\text{root mean square error, } rms = \sqrt{(y - x)(y - x)^T / N}, \tag{9}$$

$$\text{scatter index, } SI = rms / \overline{y}, \tag{10}$$

$$\text{correlation coefficient, } CC = \frac{cov(x, y)}{var(x)var(y)}. \tag{11}$$

$$\text{normalised centred rms, } RMSD = \sqrt{(y - \overline{y})(y - \overline{y})^T / (N * y_{std})}. \tag{12}$$

In addition, the complex correlation of the wind vector is determined [35] where radar wind direction is obtained from the first order Bragg ratio using a two parameter (direction and spread) short wave directional model [9,10]. This provides a complex correlation coefficient and a mean phase or direction difference.

## 3. Results

### 3.1. Statistics

For the LB site, the results of all the Sklearn methods used are summarised in Taylor Diagrams in Figure 2 using the Python SKillMetrics Taylor diagram code. In these diagrams, the black square at 1 on the $x$-axis indicates perfect agreement. The standard deviation of the radar measurements is normalised by the standard deviation in the 'truth' data set. For good agreement, this should be close to one, indicating that the radar measurements include the same variability as the 'truth', and the RMSD should be as small as possible and less than one to ensure the errors are smaller than the overall variability. For these data, the randomised tree methods, Random Forest, and its extension Extra Trees provide the best model results, although these have a tendency to over fit and produce less accurate predictions. SVR provides a reasonable model and the best prediction and the resulting scaled wind speeds for the training and test data sets are compared as 2D histograms in Figure 3.

The 2D histograms of the SVR comparisons for the Celtic Sea sites are shown in Figures 4 and 5. Although the number of cases available for testing is small in the Southern Celtic Sea, the agreement in all cases is encouraging.

SVR statistical results for all radar sites are given in Table 5. For lb, these are calculated using the actual rather than scaled speeds. Given the limitations of the data sets, these figures compare well with the scatterometer requirements: bias $< 0.5$ ms$^{-1}$ and standard deviation $< 2$ ms$^{-1}$. They are also significantly better than the existing algorithm used by Seaview Sensing, which, for these data sets, gives an rms of 3.5 to 5.0 ms$^{-1}$ and SI of 0.42 to 0.54.

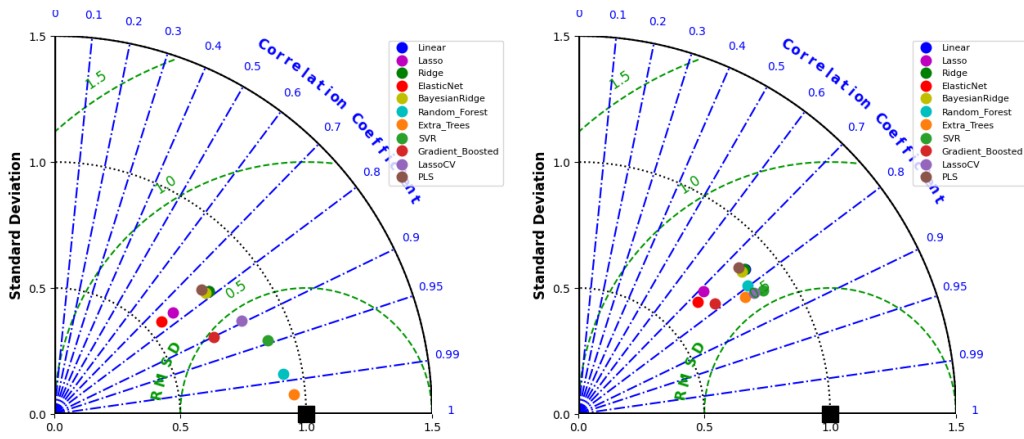

**Figure 2.** Taylor Diagrams for the Liverpool Bay training on the left and test on the right, comparing the the different methods used.

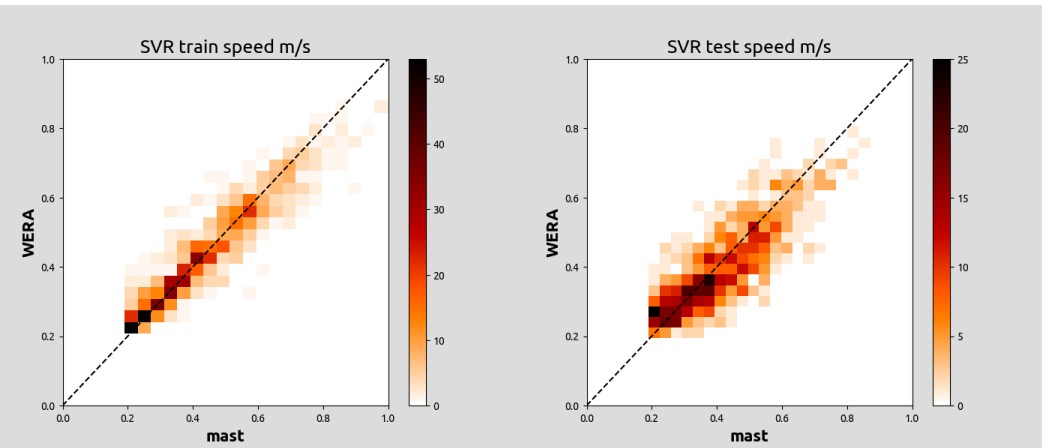

**Figure 3.** Comparison of Liverpool Bay SVR scaled wind speeds with anemometer. **Left**: training **Right**: test. Colour-coding is number of cases in each bin.

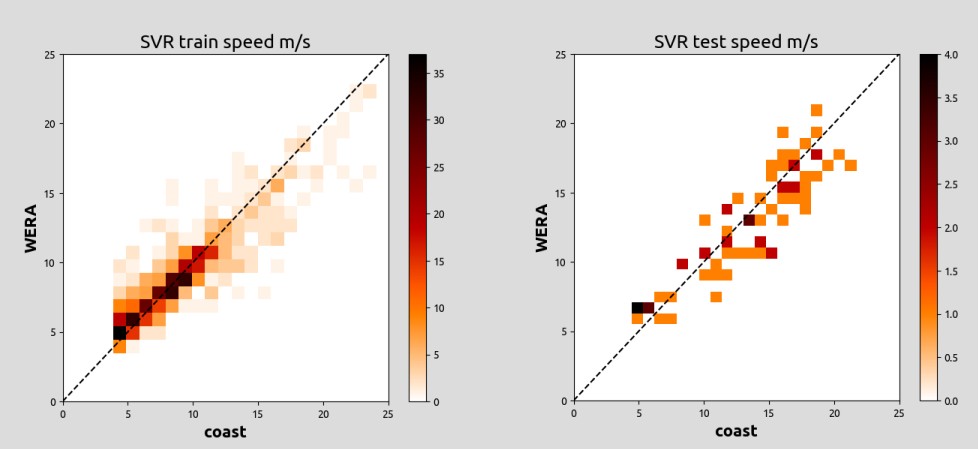

**Figure 4.** Comparison of south Celtic Sea SVR wind speeds with onshore anemometer. **Left**: training **Right**: test. Colour-coding is number of cases in each bin.

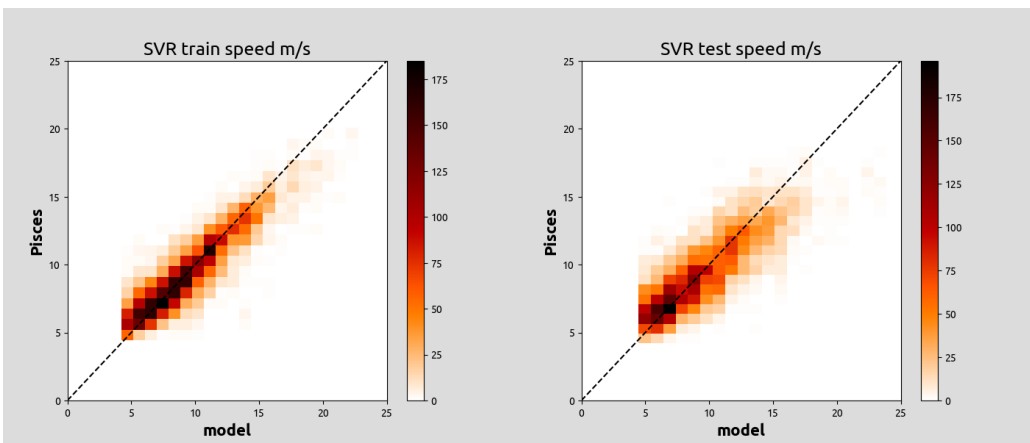

**Figure 5.** Comparison of north Celtic Sea SVR wind speeds with model speeds. **Left**: training **Right**: test. Colour-coding is number of cases in each bin.

**Table 5.** Statistics of the SVR wind speed and wind vector for train and test data sets.

| Statistic | LB | | SCS1 | | SCS2 | | NCS | |
|---|---|---|---|---|---|---|---|---|
| | **Train** | **Test** | **Train** | **Test** | **Train** | **Test** | **Train** | **Test** |
| N | 892 | 892 | 629 | 70 | 642 | 72 | 5396 | 5396 |
| bias, ms$^{-1}$ | 0.02 | 0.27 | 0.05 | 0.63 | 0.05 | 0.56 | 0.04 | 0.14 |
| MAE, ms$^{-1}$ | 0.74 | 1.18 | 1.15 | 1.68 | 1.13 | 1.58 | 1.05 | 1.45 |
| rms, ms$^{-1}$ | 1.06 | 1.5 | 1.72 | 2.05 | 1.74 | 2.01 | 1.44 | 1.92 |
| SI | 0.13 | 0.19 | 0.19 | 0.15 | 0.19 | 0.15 | 0.15 | 0.2 |
| SI$_{max}$ | 0.05 | 0.09 | 0.07 | 0.09 | 0.07 | 0.08 | 0.06 | 0.08 |
| CC | 0.95 | 0.83 | 0.9 | 0.9 | 0.91 | 0.9 | 0.9 | 0.82 |
| complex correlation coefficient | 0.94 | 0.92 | 0.93 | 0.95 | 0.93 | 0.96 | 0.91 | 0.91 |
| mean direction difference, degrees | −0.04 | 0.13 | −2.18 | −6.02 | 5.05 | −0.76 | −3.76 | −4.16 |

### 3.2. Time Series and Maps

The analysis above was confined to the locations used in the development of the ML models. In the case of lb and SCS, these were fixed locations with fixed depths, bearings relative to the radars and radio frequencies. These do not cover the parameter space associated with the coverage area of these radars, and attempts to extend them to other locations and thus provide robust wind maps were not very successful. This is discussed further in Section 4. The NCS case does provide a wider coverage of the parameter space so there is potential to assess both the performance at locations and times not used in the modelling and the quality of wind maps.

Figure 6 shows NCS wind speed and direction from the radar and Met Office wind model during the month of January 2005 at one location in the centre of the radar coverage area not included in either the training or testing data sets. The modelling was done with radar data every 6 h corresponding to the time of the Met Office wind model. In this figure, the SVR model has been applied to the hourly radar data, which reveals short-time scale variability and similar long-time scale behaviour. As was evident in the training and testing data sets (see Figures 3–5), predictions from the model have a tendency to underestimate speed at the peaks of storm events.

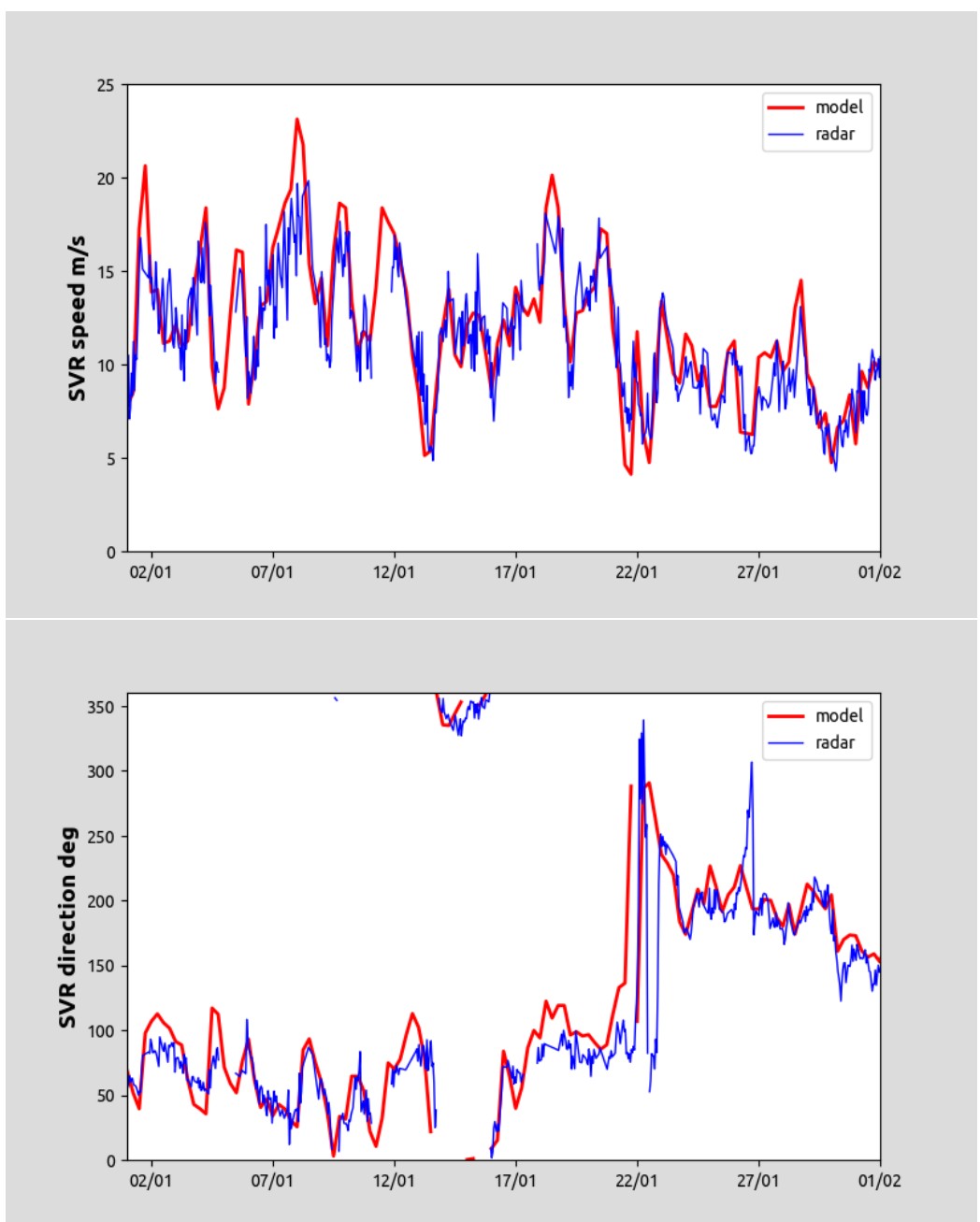

**Figure 6.** Comparison of radar and Met Office wind model wind speed (**upper**) and direction (**lower**) during January 2005.

Sample NCS maps are shown in Figure 7. This covers a period of two days of storm conditions (6 and 7 January 2005) at 6-hourly intervals, corresponding to the Met Office model data times, shown as speed-scale colour-coded circles with brown wind vectors. At two of these times, QuickSCAT data were available, and these are shown as colour-coded **x** with grey vectors. Model and QuikSCAT data are mostly similar and fairly uniform across the region. The radar data show more variability and there are some obvious errors e.g, in the centre of the region on 7 January at 6:00 a.m., perhaps linked to the underestimation of peak winds noted earlier, but the agreement is encouraging.

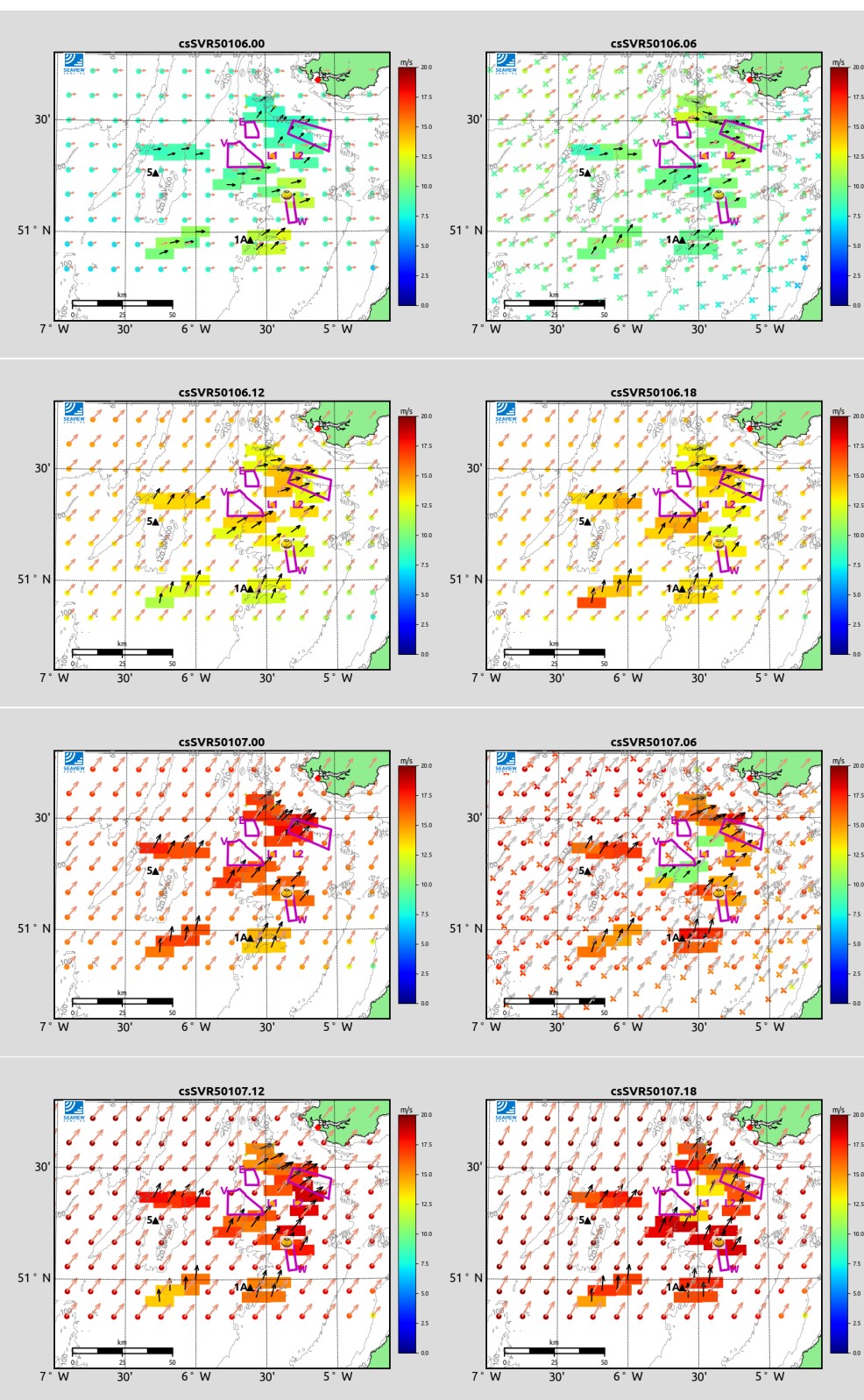

**Figure 7.** Maps of wind speed and direction for NCS every 6 h from 12:00 a.m. on 6 January 2005 (top left) to 6:00 p.m. on 7 January 2005 (bottom right). Speeds are colour-coded as shown on the scale. Large rectangles are radar cells, Met Office wind model and Quickscat are marked with circles and **x**, respectively. Black, brown, and grey arrows are radar, model and QuikSCAT directions, respectively. The magenta regions mark positions of potential offshore floating wind sites. The small buoy image marks the position of a directional waverider during the trial [7].

## 4. Discussion

The advantage of using HF radar for making wind as well as wave and current measurements is the availability of data over a wide area with high spatial and temporal resolution. The Pisces radar can now provide similar resolution to the WERA measurements, i.e., on a grid with specified cell size, rather than the coarse resolution seen in Figure 7. Satellite measurements are intermittent and may miss important events, although they provide invaluable data when assimilated into models.

The accuracies of the HF radar wind speed measurements presented in this paper and summarised in Figure 8 are probably better than any others that have appeared in the literature to date (see $SI_{max}$ in Tables 1 and 5) bearing in mind that SI would have been a more reliable comparator. They also compare well with the operational requirements for satellite scatterometer wind measurements. The feasibility of using Machine Learning methods to obtain such measurements has been demonstrated.

However, while the results are encouraging, it is clear that further work is needed. Potential sources of error are listed below.

- Lack of in situ wind measurements covering a wide enough range of radar frequencies, geometries and water depths. It is not really satisfactory to use model data since, at least in part, the purpose of providing a radar measurement is to validate or reveal errors in model data. In addition, it is necessary to establish whether the extra variability on time scales of less than six hours seen in the radar data are accurate or only reflecting statistical uncertainty in the radar measurements. We are intending to look at combining the data sets from the different sites to explore the possibility of a more generally applicable method, although this would be more satisfying if we had in-situ measured data in all cases. The use of autonomous surface vehicles to provide low-cost wind observations at multiple locations across a radar coverage area has been suggested [20]. Such an approach could be helpful in the Celtic Sea although a long trial would be needed to provide enough metocean variability. However, since the accuracies achieved in this work are so much better than the currently used algorithms, using model winds for a site specific ML model where wind speeds are likely to be useful could be a first step.
- Radar data errors. All the data sets presented here were collected over 10 years ago and improvements in radar data control and signal processing could lead to better results for this type of analysis when new radar data sets with local in-situ wind measurements become available.
- Quality control of the metocean measurements has been the subject of a lot of work [27], and all recommended filters have been applied to the data sets used here. However, better quality control of the radar feature measurements is probably required. For example, signal-to-noise limits need to be determined and estimates using single Doppler frequencies, which can have high statistical uncertainty, could be replaced with local means or centres of gravity.
- Choice of ML method and the associated hyperparameters. There is scope for more experimentation here including use of neural networks. As has been reported in Section 2.2.2, we have used the SKlearn GridSearchCV module but so far only in a limited way.
- Selection of features. It is clear that not all features have the same weight in the SVR method. There may be scope to simplify the modelling by removing features of low or negative importance. However, the relative importance of the features was found to vary for the different sites as can be seen in Figure 9 so any reduction needs to be done with care. Not surprisingly, the models obtained are different at each site so a transportable method has not yet been developed, although, as seen in Figure 8 and Table 5, accuracies are not so different.

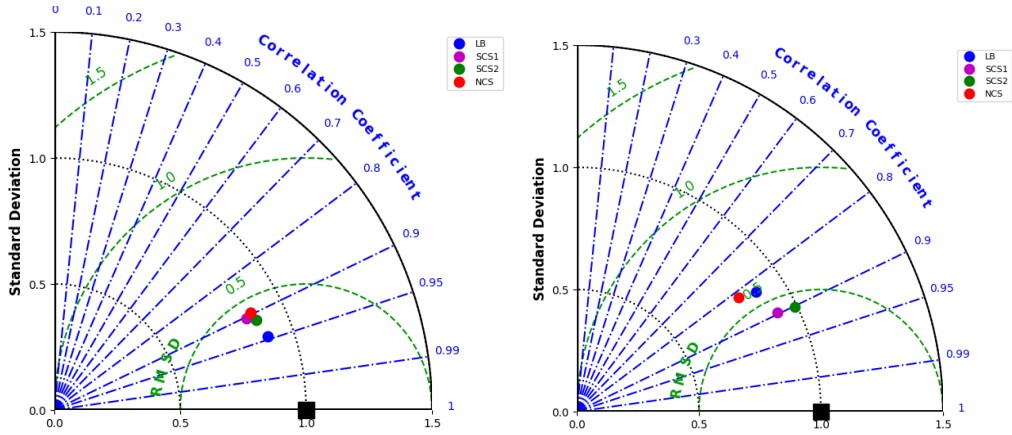

**Figure 8.** Taylor Diagrams for the training on the left and testing on the right comparing the results from the different sites.

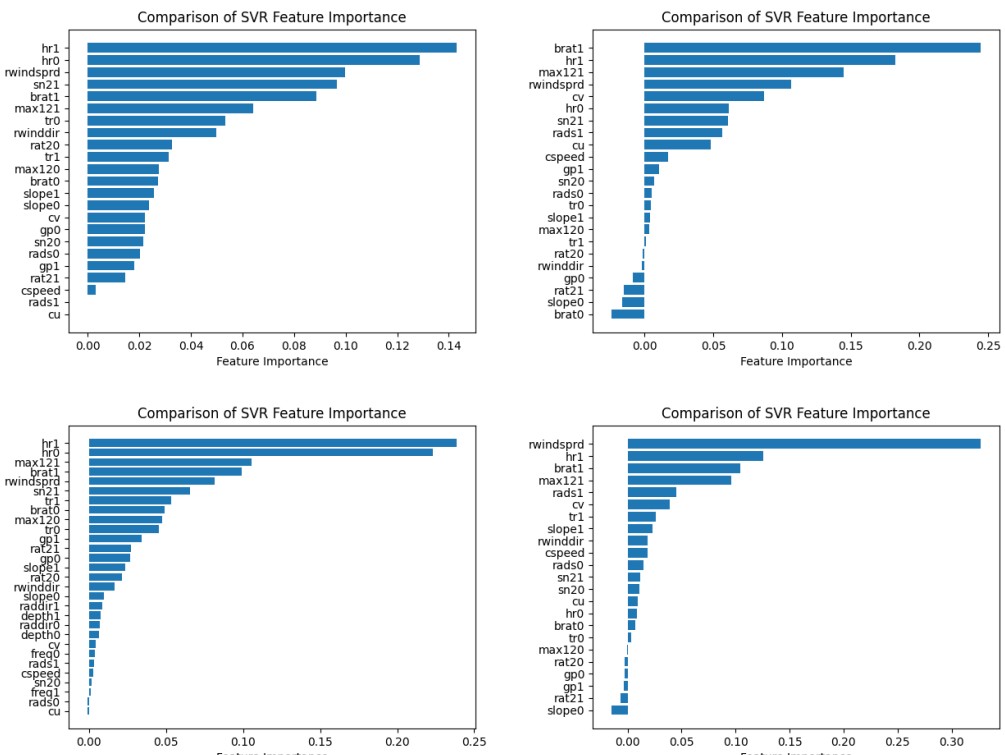

**Figure 9.** Relative importance of features and metocean parameters. The numbers 0 and 1 after a feature name refer to the estimates from the individual radars. Left column LB and NCS cases, right column SCS cases.

As previous work has shown, linear regression methods do not seem to be useful to extract wind speed from HF radar data. The wind speed signal is embedded in the Doppler power spectrum in a complicated and nonlinear way because the Doppler spectrum includes a mix of swell and wind sea impacts, which vary with radio frequency, water depth, fetch, and look direction relative to the ocean wave spectral directions and their directional spreading. This is in contrast to scatterometer and other microwave measurements which involve direct scatter from short locally generated wind waves. The ML methods used in this work show sufficient promise that a long trial with HF radar and, preferably, in-situ rather than model winds over the radar field of view, is recommended to pin down some of the issues discussed above.

**Funding:** This research received no external funding.

**Data Availability Statement:** All data used in this study are historical. No new data were created or analyzed in this study. Where data are openly available, links have been provided.

**Acknowledgments:** The radar team at National Oceanography Centre (Liverpool) provided the radar data which were collected as part of the Liverpool Bay Coastal Observatory. Amemometer data were provided by NPower renewables. The University of Plymouth radar data were provided by Daniel Conley with financial support from the Natural Environment Research Council (Grant No. NE/J004219/1). The wind data for this site is public sector information licensed under the Open Government Licence v3.0. copyright: Teignbridge DC, obtained from the Southwest Regional Coastal Monitoring Programme http://www.coastalmonitoring.org (accessed on 20 March 2022). The Pisces Celtic Sea radar data were provided by Neptune Radar and collected during a project funded by DEFRA and the Met Office who provided the wind model data. QuikSCAT Level 2B Ocean Wind Vectors in 12.5 km Slice Composites Version 4.1 were downloaded from https://earthdata.nasa.gov/ (accessed on 20 March 2022).

**Conflicts of Interest:** Lucy Wyatt is one of the founders and part-owners of Seaview Sensing Ltd. She has a history of publications that demonstrate objective scientific research. The company may eventually benefit from the research results presented here when a commercial product has been finalised.

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
