# Peer review of "Progress towards an HF Radar Wind Speed Measurement Method Using Machine Learning"

_remotesensing, doi:10.3390/rs14092098_

Round 1

Reviewer 1 Report

The manuscript is dedicated to the sea wind speed measurement by the HF radar using machine learning. The sea wind speed measurement methods proposed by now in the world have been reviewed and an alternative approach using machine learning methods has been developed. The results obtained are new and valuable for the field. The paper is structured and written well. It can be published as is but some small corrections are desirable.

Corrections suggested.

  1. Table 1. Column “method”. Better to add the name of the method used for [16] and [20].

  1. Please, insert blank between the value and its dimension.

  1. Table 3. Column “Reason”. Better to write the reason by words then to provide only references [30] and [31].

  1. Please, provide left and right figure description in the figure caption for Figure 3, 4, and 5.

  1. Use e.g. Figure 1 instead of Fig.1. in the text.

  1. Please, abbreviate journal names in the journal references. Please, provide DOIs for the references as much as possible.

Author Response

Thank you for your comments. I hope the changes made are clear in the edited version of the revised manuscript. My responses to your specific comments are

  1. Information added.
  2. I hope I found all of these and corrected them.
  3. Information added.
  4. This is done. The figures have been changed to address a comment from reviewer 2 but the details are the same.
  5. I hope I found all of these and corrected them.
  6. Abbreviations and DOIs done where possible.

Reviewer 2 Report

I have a few suggestions and comments for the manuscript:

1) In Table 1 please include a shorthand for the citation (rather than number), and also I think it would be possible to estimate the scatter index (SI) for the table if the max wind value is used rather than the mean. This would also help assess the relative accuracy of the results presented here.

2) The requirements of the Carbon Trust on line 70 seem to be quite extreme. Would they get this between similar instruments separated by a few meters?

3)Line 98: Suggest consistent  use of geographic names, since 'Cornish coast' and North Celtic Sea are not obviously related (or show us a map I suppose).

4) Line 104 I think this should be "90:10 into training and test data sets". Also, I question the value of adding these results but the author does provide the appropriate caveats. However "Using few data in the training set did not give satisfactory results" is probably not a valid justification.

5) Line 128: please quantify SNR values that are "too low" eg less than 10?

6) Is metocean a real word?

7) Line 160-164: Is this section referring to second order Bragg?

Figs 3 and 4 (and elsewhere) I suggest consistent use of the words 'training' and 'test' throughout, including the figures (which use 'predict') for example

8) Line 244 and figure 6 use 'wind model' instead of model please

9) Line 267: It seems to me that the results in this manuscript are similar to others, but producing SI for all of them as suggested above would make it more clear.

Overall a very nice paper.

Author Response

Thank you for your comments. I hope the changes made are clear in the edited version of the revised manuscript. My responses to your specific comments are

  1. Extra details added to table. I have calculated SI using max wind speed as suggested and added to the table and commented on this on li 58-62.
  2. These requirements are only for lidar vs mast comparisons as mentioned. It seems to work in that context but we cannot achieve that. I only included this for background.
  3. Changed Cornish coast to South Celtic Sea. li 102.
  4. Thanks for that - 'test' added. I have also extended the sentence (li 109-111)referred to to clarify.
  5. I have added the 8dB threshold used on line 134.
  6. Metocean is indeed a real word - a term for wind, wave and current measurements.
  7. Yes, thanks I have added this at li 170.

Training and Test have been used in most places now including on those figures. The use of the words model and prediction is still there in a couple of places for clarity.

8. I have used Met Office wind model for complete clarity.

9. SI using max wind speed also added to table 5 where it clear the new results produce smaller number than those in Table 1. li 275-276

Reviewer 3 Report

I think this manuscript is well written and the results are very encouraging. The author has made a further step towards operational measurements of the wind speed with an HF radar. 

My only suggestion is that, if possible, could you add some comparisons with the non-machine learning method, i.e., model inversion?  

Author Response

Thank you for your comments. Some of the results in Table 1 are from non-machine learning methods. I also included a comparison with the inverse wind wave model for these specific data sets in lines 239-241. I hope these address your suggestion.